# Sparse Steerable Convolutions: An Efficient Learning of SE(3)-Equivariant Features for Estimation and Tracking of Object Poses in 3D Space

**Jiehong Lin**[1,2]*, **Hongyang Li**[1]*, **Ke Chen**[1,3], **Jiangbo Lu**[4], **Kui Jia**[1,3,5]†
[1]South China University of Technology. [2]DexForce Co. Ltd.
[3]Peng Cheng Laboratory. [4]SmartMore Technology Co. Ltd. [5]Pazhou Laboratory.
{lin.jiehong, eeli.hongyang}@mail.scut.edu.cn
jiangbo.lu@gmail.com, {chenk, kuijia}@scut.edu.cn

## Abstract

As a basic component of SE(3)-equivariant deep feature learning, steerable convolution has recently demonstrated its advantages for 3D semantic analysis. The advantages are, however, brought by expensive computations on dense, volumetric data, which prevent its practical use for efficient processing of 3D data that are inherently sparse. In this paper, we propose a novel design of *Sparse Steerable Convolution (SS-Conv)* to address the shortcoming; SS-Conv greatly accelerates steerable convolution with sparse tensors, while strictly preserving the property of SE(3)-equivariance. Based on SS-Conv, we propose a general pipeline for precise estimation of object poses, wherein a key design is a Feature-Steering module that takes the full advantage of SE(3)-equivariance and is able to conduct an efficient pose refinement. To verify our designs, we conduct thorough experiments on three tasks of 3D object semantic analysis, including instance-level 6D pose estimation, category-level 6D pose and size estimation, and category-level 6D pose tracking. Our proposed pipeline based on SS-Conv outperforms existing methods on almost all the metrics evaluated by the three tasks. Ablation studies also show the superiority of our SS-Conv over alternative convolutions in terms of both accuracy and efficiency. Our code is released publicly at https://github.com/Gorilla-Lab-SCUT/SS-Conv.

## 1 Introduction

SE(3)-equivariant deep networks [20, 25, 7] have shown the promise recently in some tasks of 3D semantic analysis, among which 3D Steerable CNN [25] is a representative one. 3D Steerable CNNs employ steerable convolutions (termed as *ST-Conv*) to learn pose-equivariant features in a layer-wise manner, thus preserving the pose information of the 3D input. Intuitively speaking, for a layer of ST-Conv, any SE(3) transformation $(\boldsymbol{r}, \boldsymbol{t})$ applied to its 3D input would induce a pose-synchronized transformation to its output features, where $\boldsymbol{r} \in$ SO(3) stands for a rotation and $\boldsymbol{t} \in \mathbb{R}^3$ for a translation. Fig. 1 (a) gives an illustration where given an SE(3) transformation of the input, the locations at which feature vectors are defined are rigidly transformed with respect to $(\boldsymbol{r}, \boldsymbol{t})$, and the feature vectors themselves are also rotated by $\rho(\boldsymbol{r})$ ($\rho(\boldsymbol{r})$ is a representation of rotation $\boldsymbol{r}$). This property of SE(3)-equivariance enables the steerability of feature space. For example, without transforming the input, SE(3) transformation can be directly realized by steering in the feature space. To produce steerable features, ST-Conv confines its feature domain to regular grids of 3D volumetric

---

*Equal contribution

†Corresponding author

35th Conference on Neural Information Processing Systems (NeurIPS 2021).

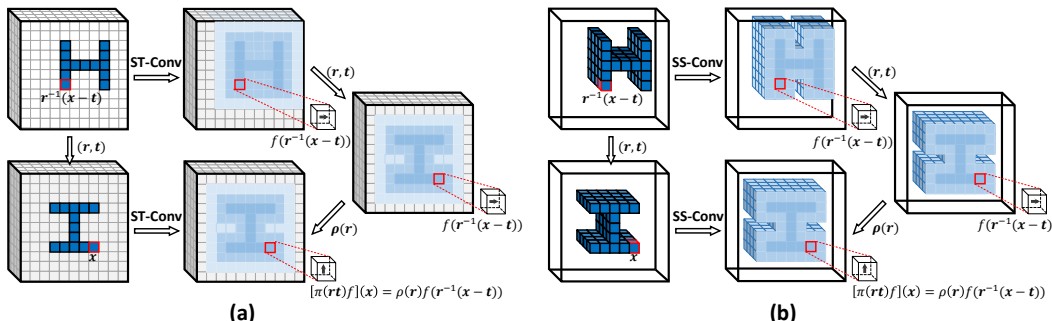

Figure 1: An illustration of SE(3)-equivariance achieved by (a) STeerable Convolution (ST-Conv) based on dense tensors, and (b) our Sparse Steerable Convolution (SS-Conv) based on sparse tensors, where arrows defined on the 3D fields denote vector-formed, oriented features. Best view in the electronic version.

data; it can thus be conveniently supported by 3D convolution routines. This compatibility with 3D convolutions eases the implementation of ST-Conv, but at the sacrifice of efficiently processing 3D data (e.g., point clouds) that are typically irregular and sparse; consequently, ST-Conv is still less widely used in broader areas of 3D semantic analysis.

In this paper, we propose a novel design of *Sparse Steerable Convolution (SS-Conv)* to address the aforementioned shortcoming faced by ST-Conv. SS-Conv can greatly accelerate steerable convolutions with sparse tensors, while strictly preserving the SE(3)-equivariance in feature learning; Fig. 1 (b) gives the illustration. To implement SS-Conv, we construct convolutional kernels as linear combinations of basis ones based on spherical harmonics, which satisfy the rotation-steerable constraint of SE(3)-equivariant convolutions [25], and implement the convolution as matrix-matrix multiply-add operations on GPUs only at active sites, which are recorded along with their features as sparse tensors.

Although SE(3)-equivariant feature learning is widely used in 3D object recognition, its potentials for other tasks of 3D semantic analysis have not been well explored yet. In this work, we make the attempt to apply our proposed SS-Conv to object pose estimation in 3D space. To this end, we propose a general pipeline based on SS-Conv, which stacks layers of SS-Conv as the backbone, and decodes object poses directly from the learned SE(3)-equivariant features. A novel Feature-Steering module is also designed into the pipeline to support iterative pose refinement, by taking advantage of the steerability of the learned features. We conduct thorough experiments on three tasks of pose-related, 3D object semantic analysis, including instance-level 6D pose estimation, category-level 6D pose and size estimation, and category-level 6D pose tracking. Our proposed pipeline based on SS-Conv outperforms existing methods on almost all the metrics evaluated by the three tasks; the gaps are clearer in the regimes of high-precision pose estimation. Ablation studies also show the superiority of our SS-Conv over alternative convolutions in terms of both accuracy and efficiency.

## 2 Related Work

**SE(3)-Equivariant Representation Learning** SE(3)-equivariance is an important property in 3D computer vision. In earlier works, researchers ease the problem by focusing on SO(3)-equivariance, and design Spherical CNNs [6, 5] by stacking SO(3)-equivariant spherical convolutions which are implemented in the spherical harmonic domain. Recently, a series of works [20, 25, 7] build deep SE(3)-equivariant networks based on steerable kernels, which are parameterized as linear combinations of basis kernels. Thomas *et al.* firstly propose Tensor Field Network (TFN) [20] to learn SE(3)-equivariant features on irregular point clouds, and later, Fuchs *et al.* present SE(3)-Transformer, which extends TFN with attention mechanism. However, those networks working on point clouds are required to compute kernels with respect to different input points inefficiently. To tackle this problem, 3D steerable convolution (ST-Conv) [25] is proposed to work on regular volumetric data, so that basis kernels with respect to regular grids could be pre-computed; however, it still encounters challenging computational demands due to the ignorance of data sparsity. Compared to the above methods, our proposed sparse steerable convolution aims at efficient SE(3)-equivariant

representation learning for volumetric data, which is realized with sparse tensors to accelerate the computation.

**Estimation and Tracking of Object Poses in 3D Space** In the context of pose estimation, instance-level 6D pose estimation is a classical and well-developed task, for which a body of works are proposed. These works can be broadly categorized into three types: i) template matching [12] by constructing templates to search for the best matched poses; ii) 2D-3D correspondence methods [1, 14, 16, 19, 17], which establish 2D-3D correspondence via 2D keypoint detection [19, 17] or dense 3D coordinate predictions [1, 14, 16], followed by a PnP algorithm to obtain the target pose; iii) direct pose regression [26, 13, 23] via deep networks. Recently, a more challenging task of category-level 6D pose and size estimation is formally introduced in [24], aiming at estimating poses of 3D unknown objects with respect to a categorical normalized object coordinate space (NOCS). The early works [24, 21] focus on regression of NOCS maps, and the poses can be obtained by aligning NOCS maps with the observed depth maps. Later, methods of direct pose regression are proposed thanks to the special designs of fusion of pose-dependent and pose-independent features [2], decoupled rotation mechanism [4], or dual pose decoders [15]. Motivated by [24], Wang *et. al* propose the task of category-level 6D pose tracking, aiming for the small change of object poses between two adjacent frames in a sequence; they also present 6-PACK, a pose tracker estimating the change of poses by matching keypoints of two frames.

## 3 Sparse Steerable Convolutional Neural Network

### 3.1 Background

**3D Convolution** A conventional 3D convolution can be formulated as follows:

$$f_{n+1}(\boldsymbol{x}) = [\kappa \star f_n](\boldsymbol{x}) = \int_{\mathbb{R}^3} \kappa(\boldsymbol{x} - \boldsymbol{y}) f_n(\boldsymbol{y}) d\boldsymbol{y}, \tag{1}$$

where $f_n(\boldsymbol{x}) \in \mathbb{R}^{K_n}$, $f_{n+1}(\boldsymbol{x}) \in \mathbb{R}^{K_{n+1}}$, and $\kappa : \mathbb{R}^3 \to \mathbb{R}^{K_{n+1} \times K_n}$ is a continuous learnable kernel.

**SE(3)-Equivariance** Given a transformation $\pi_n(\boldsymbol{g}) : \mathbb{R}^{K_n} \to \mathbb{R}^{K_n}$ for a 3D rigid motion $\boldsymbol{g} \in$SE(3), a 3D convolution in Eq. (1) is SE(3)-equivariant if there exists a transformation $\pi_{n+1}(\boldsymbol{g}) : \mathbb{R}^{K_{n+1}} \to \mathbb{R}^{K_{n+1}}$ such that

$$[\pi_{n+1}(\boldsymbol{g}) f_{n+1}](\boldsymbol{x}) = [\kappa \star [\pi_n(\boldsymbol{g}) f_n]](\boldsymbol{x}). \tag{2}$$

Such an SE(3)-equivariant convolution is *steerable*, since the feature $f_{n+1}(\boldsymbol{x})$ can be steered by $\pi_{n+1}(\boldsymbol{g})$ in the feature space [25].

In general, the transformation $\pi_n(\boldsymbol{g})$ is a group representation of SE(3), which satisfies $\pi_n(\boldsymbol{g_1}\boldsymbol{g_2}) = \pi_n(\boldsymbol{g_1})\pi_n(\boldsymbol{g_2})$. If $\boldsymbol{g}$ is decomposed into a 3D rotation $\boldsymbol{r} \in$ SO(3) and a 3D translation $\boldsymbol{t} \in \mathbb{R}^3$, written as $\boldsymbol{g} = \boldsymbol{tr}$, $\pi_n(\boldsymbol{g})$ can be defined as follows:

$$[\pi_n(\boldsymbol{g}) f_n](\boldsymbol{x}) = [\pi_n(\boldsymbol{tr}) f_n](\boldsymbol{x}) := \rho_n(\boldsymbol{r}) f_n(\boldsymbol{r}^{-1}(\boldsymbol{x} - \boldsymbol{t})), \tag{3}$$

where $\rho_n(\boldsymbol{r}) : \mathbb{R}^{K_n} \to \mathbb{R}^{K_n}$ is an SO(3) representation. The illustration is given in Fig. 1.

**Rotation-Steerable Constraint** To guarantee SE(3)-equivariance in Eq. (2), it can be derived that the kernel $\kappa$ of 3D convolution must be rotation-steerable [25], which satisfies the following constraint:

$$\kappa(\boldsymbol{rx}) = \rho_{n+1}(\boldsymbol{r})\kappa(\boldsymbol{x})\rho_n(\boldsymbol{r})^{-1}. \tag{4}$$

**Irreducible Feature** $\rho_n(\boldsymbol{r})$ is an SO(3) representation, which can be decomposed into $F_n$ *irreducible representations* as follows:

$$\rho_n(\boldsymbol{r}) = \boldsymbol{Q}^T [\bigoplus_{i=0}^{F_n} D^{l_i}(\boldsymbol{r})] \boldsymbol{Q}, \tag{5}$$

where $\boldsymbol{Q}$ is a $K_n \times K_n$ change-of-basis matrix, $D^{l_i}(\boldsymbol{r})$ is the $(2l_i + 1) \times (2l_i + 1)$ irreducible Wigner-D matrix [8] of order $l_i$ ($l_i = 0, 1, 2, ...$), and $\bigoplus$ represents block-diagonal construction of $\{D^{l_i}(\boldsymbol{r})\}$, so that $K_n = \sum_{i=0}^{F_n} 2l_i + 1$. Based on Eq. (3), $f_n(\boldsymbol{x})$ can be constructed by stacking $F_n$ *irreducible features* $\{f_n^i(\boldsymbol{x}) \in \mathbb{R}^{2l_i+1}\}$; each $f_n^i(\boldsymbol{x})$ is associated with a $D^{l_i}(\boldsymbol{r})$. When $l_i = 0$, $D^0(\boldsymbol{r}) = 1$, so that $f_n^i(\boldsymbol{x})$ is a scalar invariant to any rotation; when $l_i > 0$, $f_n^i(\boldsymbol{x})$ is a vector which can be rotated by $D^{l_i}(\boldsymbol{r})$.

## 3.2 Sparse Steerable Convolution

*3D STeerable Convolution* (**ST-Conv**) enjoys the property of SE(3)-equivariance; however, as discussed in Sec. 1, it suffers from heavy computations as conventional 3D convolution does. Motivated by recent success of *SParse Convolution* (**SP-Conv**) [9], we propose a novel design of *Sparse Steerable Convolution* (**SS-Conv**) with sparse tensors, which takes the natural sparsity of 3D data into account, while strictly keeping the steerability of features.

Specifically, assuming $\kappa$ is a discretized, $s \times s \times s$, cubic kernel with grid sites $\boldsymbol{S} = \{-\frac{s-1}{2}, ..., -1, 0, 1, ..., \frac{s-1}{2}\}^3$ ($s$ is an odd), our proposed SS-Conv can be formulated as follows:

$$f_{n+1}(\boldsymbol{x}) = [\kappa \star f_n](\boldsymbol{x}) = \begin{cases} \sum_{\boldsymbol{x}-\boldsymbol{y}\in\boldsymbol{S},\sigma_n(\boldsymbol{y})=1} \kappa(\boldsymbol{x}-\boldsymbol{y})f_n(\boldsymbol{y}), & \text{if } \sigma_{n+1}(\boldsymbol{x}) = 1 \\ \boldsymbol{0}, & \text{if } \sigma_{n+1}(\boldsymbol{x}) = 0 \end{cases} \quad (6)$$

$$s.t. \quad \forall \boldsymbol{r} \in SO(3), \kappa(\boldsymbol{r}\boldsymbol{x}) = \rho_{n+1}(\boldsymbol{r})\kappa(\boldsymbol{x})\rho_n(\boldsymbol{r})^{-1},$$

where $\sigma_n(\boldsymbol{x})$ represents the state of site $\boldsymbol{x}$ in the feature space $\mathbb{R}^{K_n}$.[1] $\sigma_n(\boldsymbol{x}) = 0$ denotes an inactive state at $\boldsymbol{x}$, where $f_n(\boldsymbol{x})$ is in its ground state; when $f_n(\boldsymbol{x})$ is beyond the ground state, this site would be activated as $\sigma_n(\boldsymbol{x}) = 1$. In SS-Conv, we set the ground state as a zero vector.

Compared with ST-Conv, our sparse version is accelerated in two ways: i) convolutions are conducted at activated output sites, not on the whole 3D volume, where the number of active sites only takes a small proportion; ii) in the receptive field of each activated output site, only active input features are convolved. For these purposes, we represent the input and output features as sparse tensors $(\boldsymbol{H}_n, \boldsymbol{F}_n)$ and $(\boldsymbol{H}_{n+1}, \boldsymbol{F}_{n+1})$, respectively. $\boldsymbol{H}_n$ and $\boldsymbol{H}_{n+1}$ are hash tables recording the coordinates of active sites only, while $\boldsymbol{F}_n$ and $\boldsymbol{F}_{n+1}$ are feature matrices. For a sparse tensor, its hash table and feature matrix correspond to each other row-by-row; that is, if $r_{n+1,\boldsymbol{x}}$ is the row number of $\boldsymbol{x}$ in $\boldsymbol{H}_{n+1}$, then $\boldsymbol{F}_{n+1}[r_{n+1,\boldsymbol{x}}] = f_{n+1}(\boldsymbol{x})$.

In this respect, the goal of SS-Conv is to convolve $(\boldsymbol{H}_n, \boldsymbol{F}_n)$ with $\kappa$ to obtain $(\boldsymbol{H}_{n+1}, \boldsymbol{F}_{n+1})$, which can be implemented in three steps: i) **Rotation-Steerable Kernel Construction** (cf. 3.2.1) for generation of $\kappa$, ii) **Site State Definition** (cf. 3.2.2) for the output hash table $\boldsymbol{H}_{n+1}$, and **Sparse Convolutional Operation** (cf. 3.2.3) for the output feature matrix $\boldsymbol{F}_{n+1}$. We will introduce the detailed implementations shortly.

### 3.2.1 Rotation-Steerable Kernel Construction

The key to satisfy the rotation-steerable constraint (4) is to control the angular directions of feature vectors, and a recent research shows that spherical harmonics $Y^J = \{Y_j^J\}_{j=-J}^J$ give the unique and complete solution [25]. Linear combination of the basis kernels based on spherical harmonics produces the rotation-steerable convolutional kernel $\kappa$.

For simplicity, we firstly consider both input and output features as individual irreducible ones of orders $l$ and $k$, respectively; the kernel $\kappa^{kl} : \mathbb{R}^3 \to \mathbb{R}^{(2k+1)\times(2l+1)}$ is parameterized as a linear combination of basis kernels $\kappa^{kl,Jm} : \mathbb{R}^3 \to \mathbb{R}^{(2k+1)\times(2l+1)}$:

$$\kappa^{kl}(\boldsymbol{x}) = \sum_{J=|k-l|}^{k+l} \sum_m w^{kl,Jm} \kappa^{kl,Jm}(\boldsymbol{x}), \quad (7)$$

where

$$\kappa^{kl,Jm}(\boldsymbol{x}) = \sum_{j=-J}^J \varphi^m(\|\boldsymbol{x}\|) Y_j^J\left(\frac{\boldsymbol{x}}{\|\boldsymbol{x}\|}\right) \boldsymbol{Q}_j^{kl}. \quad (8)$$

$\boldsymbol{Q}_j^{kl}$ is a $(2k+1) \times (2l+1)$ change-of-basis matrix, also known as Clebsch-Gordan coefficients, and $\varphi^m$ is a continuous Gaussian radial function: $\varphi^m(\|\boldsymbol{x}\|) = e^{-\frac{1}{2}(\|\boldsymbol{x}\|-m)^2/\epsilon^2}$. In the basis kernel $\kappa^{kl,Jm}$ (8), $Y^J$ controls the angular direction, while $\varphi^m$ controls the radial one; then $\{\kappa^{kl,Jm}\}$ are linearly combined by learnable coefficients $\{w^{kl,Jm}\}$ as in Eq. (7) to further adjust the radial

---

[1]We only discuss convolutions (stride = 1) in our paper, since as pointed out in [25], convolutions (stride > 1) damage the smoothness of features and break the properties of equivariance. For feature downsampling, we follow [25] and use a combination of a convolution (stride = 1) with an average pooling.

direction, which is the only degree of freedom in the process of optimization. Accordingly, the angular direction is totally controlled by $Y^J$, such that the rotation-steerable constraint is strictly followed. In addition, the total number of learnable parameters in Eq. (7) is $M[2min(k,l) + 1]$ ($M$ is the number of selected $\{m\}$), which is, in practice, marginally less than that of conventional 3D convolution, which has $(2k + 1)(2l + 1)$ parameters.

Finally, assuming that the input and output features are stacked irreducible features, whose orders are $\{l_1, \cdots, l_{F_n}\}$ and $\{k_1, \cdots, k_{F_{n+1}}\}$ respectively, the rotation-steerable kernel of SS-Conv can be constructed as follows:

$$\kappa(\boldsymbol{x}) = \begin{bmatrix} \kappa^{k_1 l_1}(\boldsymbol{x}) & \cdots & \kappa^{k_1 l_{F_n}}(\boldsymbol{x}) \\ \vdots & \ddots & \vdots \\ \kappa^{k_{F_{n+1}} l_1}(\boldsymbol{x}) & \cdots & \kappa^{k_{F_{n+1}} l_{F_n}}(\boldsymbol{x}) \end{bmatrix}, \tag{9}$$

with the size of $K_{n+1} \times K_n$, where $K_{n+1} = \sum_{i=1}^{F_{n+1}} 2k_{F_{n+1}} + 1$ and $K_n = \sum_{i=1}^{F_n} 2l_{F_n} + 1$.

### 3.2.2 Site State Definition

The key to enable the efficiency of SS-Conv lies in the definition of site state. In general, for an output grid site $\boldsymbol{x}$, if any of input sites in its receptive field are active, this site will be activated, and convolution at this site will be conducted; otherwise, this site will keep inactive, meaning that its feature will be directly set as a zero vector (representing the ground state) without convolutional operation. We formulate the above definition of state at site $\boldsymbol{x}$ as follows:

$$\sigma_{n+1}(\boldsymbol{x}) = \begin{cases} 1, & \text{if } \exists\, \boldsymbol{y} \in \boldsymbol{H}_n \text{ and } \boldsymbol{x} - \boldsymbol{y} \in \boldsymbol{S} \\ 0. & \text{others} \end{cases} \tag{10}$$

The output hash table $\boldsymbol{H}_{n+1}$ is then generated as $\{\boldsymbol{x} : \sigma_{n+1}(\boldsymbol{x}) = 1\}$.

The number of active sites defined in (10) will increase layer-by-layer, enabling long-range message transfer. However, if dozens or even hundreds of convolutions are stacked, the rapid growth rate of active sites would result in heavy computational burden and the so-called *"submanifold dilation problem"* [9]. To alleviate this problem, we follow [9] and consider another choice of state definition in our SS-Conv, which keeps the output state consistent with the input one at a same grid site, *i.e.*, $\sigma_{n+1}(\boldsymbol{x}) = \sigma_n(\boldsymbol{x})$, such that $\boldsymbol{H}_{n+1} = \boldsymbol{H}_n$. This kind of SS-Conv without dilation makes it possible to construct a deep but efficient network for sparse volumetric data, and we term it as "*Submanifold SS-Conv*".

In practice, we mix general SS-Convs and Submanifold SS-Convs in an alternating manner to achieve high accuracy and efficiency.

### 3.2.3 Sparse Convolutional Operation

After obtaining $\boldsymbol{H}_{n+1}$, the next target is to compute the values of $\boldsymbol{F}_{n+1}$. Specifically, we firstly initialize $\boldsymbol{F}_{n+1}$ to zeros; then the feature vectors in $\boldsymbol{F}_{n+1}$ are updated via the following algorithm:

---

**ALGORITHM 1**: Sparse Steerable Convolution

---

**Input**: $(\boldsymbol{H}_n, \boldsymbol{F}_n), (\boldsymbol{H}_{n+1}, \boldsymbol{F}_{n+1}), \{\kappa(\boldsymbol{s}) : \boldsymbol{s} \in \boldsymbol{S}\}$
**Output**: $(\boldsymbol{H}_{n+1}, \boldsymbol{F}_{n+1})$

| | | |
|---|---|---|
| 1: | $\boldsymbol{R} = \{\boldsymbol{R}_{\boldsymbol{s}} = \varnothing : \boldsymbol{s} \in \boldsymbol{S}\}$ | // Initialize the rule book $\boldsymbol{R}$. |
| 2: | **for** $\boldsymbol{x}$ in $\boldsymbol{H}_{n+1}$ **do** | // Construct the rule book $\boldsymbol{R}$. |
| 3: |     **for** $\boldsymbol{y}$ in $\boldsymbol{H}_n$ **do** | |
| 4: |         **if** $\boldsymbol{s} = \boldsymbol{x} - \boldsymbol{y} \in \boldsymbol{S}$: | |
| 5: |             Append $(r_{n+1,\boldsymbol{x}}, r_{n,\boldsymbol{y}})$ to $\boldsymbol{R}_{\boldsymbol{s}}$. | // $r_{n+1,\boldsymbol{x}}$ is the row number of $\boldsymbol{x}$ in $\boldsymbol{H}_{n+1}$. |
| 6: |     **end for** | // $r_{n,\boldsymbol{y}}$ is the row number of $\boldsymbol{y}$ in $\boldsymbol{H}_n$. |
| 7: | **end for** | |
| 8: | **for** $\boldsymbol{R}_{\boldsymbol{s}}$ in $\boldsymbol{R}$ **do** | // Update $\boldsymbol{F}_{n+1}$. |
| 9: |     **for** $(r_{n+1,\boldsymbol{x}}, r_{n,\boldsymbol{y}})$ **in** $\boldsymbol{R}_{\boldsymbol{s}}$ **do** | |
| 10: |         $\boldsymbol{F}_{n+1}[r_{n+1,\boldsymbol{x}}] \Leftarrow \boldsymbol{F}_{n+1}[r_{n+1,\boldsymbol{x}}] + \kappa(\boldsymbol{s}) \times \boldsymbol{F}_n[r_{n,\boldsymbol{y}}]$ | |
| 11: |     **end for** | |
| 12: | **end for** | |

---

This process can be divided into two substeps. The first one is to construct a rule book $\boldsymbol{R} = \{\boldsymbol{R}_{\boldsymbol{s}} : \boldsymbol{s} \in \boldsymbol{S}\}$, where an active output site $\boldsymbol{x}$ is paired with an active input $\boldsymbol{y}$ in each $\boldsymbol{R}_{\boldsymbol{s}}$, if $\boldsymbol{x} - \boldsymbol{y} = \boldsymbol{s}$. The

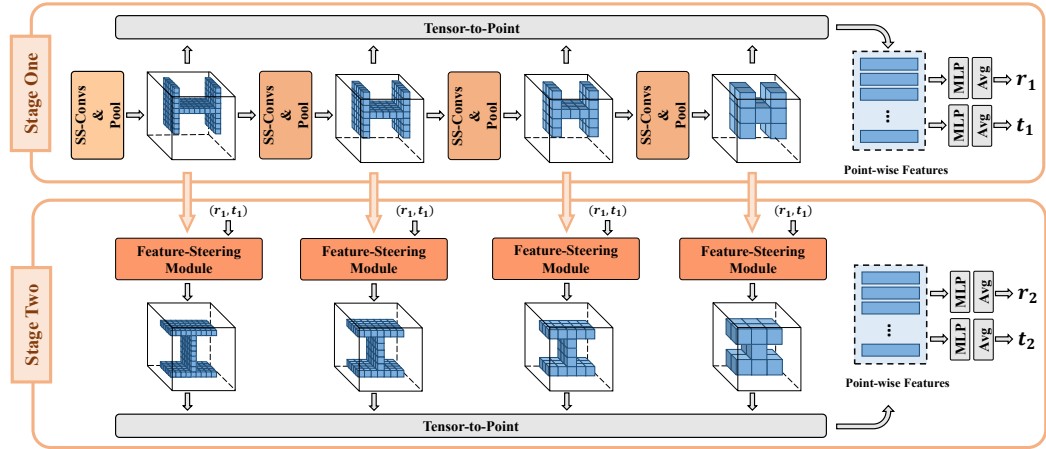

Figure 2: An illustration of network architecture for instance-level 6D object pose estimation.

second one is to update $\boldsymbol{F}_{n+1}$ according to the paired relationships recorded in $\boldsymbol{R}$; for example, if the paired relationship of output $\boldsymbol{x}$ and input $\boldsymbol{y}$ is recorded in $\boldsymbol{R}_{\boldsymbol{s}}$, the current $f_{n+1}(\boldsymbol{x})$ will be updated by adding the multiplication of $f_n(\boldsymbol{y})$ and $\kappa(\boldsymbol{s})$. In this process, the construction of $\boldsymbol{R}$ is very critical, which helps to implement the second substep by matrix-matrix multiply-add operations on GPUs efficiently.

### 3.3 Normalization and Activation

As conventional CNNs do, SS-Convs are also followed by normalization and activation, *i.e.*, *Activation*($Norm([\kappa \star f_n](\boldsymbol{x}))$). Those operations of normalization and activation are required to be specially designed, not to break the SE(3)-equivariance of features. Since each SE(3)-equivariant feature is formed by stacking irreducible ones, without loss of generality, we take as an example an irreducible feature $\boldsymbol{f}(\boldsymbol{x})$ with order $l$, so that the normalization can be formulated as follows:

$$Norm(f(\boldsymbol{x})) = \begin{cases} (f(\boldsymbol{x}) - \mathrm{E}[f(\boldsymbol{x})])/\sqrt{\mathrm{Var}[f(\boldsymbol{x})] + \epsilon'}, & l = 0 \\ f(\boldsymbol{x})/\sqrt{\mathrm{E}[\|f(\boldsymbol{x})\|^2] + \epsilon'}, & l > 0 \end{cases} \tag{11}$$

where $\mathrm{E}[\cdot]$ and $\mathrm{Var}[\cdot]$ are population mean and variance, respectively. $\epsilon'$ is a very small constant. For the activation of $f(\boldsymbol{x})$, if $l = 0$, *ReLU* can be chosen to increase non-linearity; if $l > 0$, we follow [25] and multiply to $f(\boldsymbol{x})$ a scalar, which is learned by a SS-Conv and applied to the *Sigmoid* function:

$$Activation(f(\boldsymbol{x})) = \begin{cases} ReLU(f(\boldsymbol{x})), & l = 0 \\ Sigmoid([\kappa^{0l} \star f](\boldsymbol{x}))f(\boldsymbol{x}). & l > 0 \end{cases} \tag{12}$$

The above normalization and activation operations are both SE(3)-equivariant, since a feature vector multiplying any scalar keeps its equivariance; when applying them to features formed by numerous irreducible ones, we treat each irreducible member individually to ensure the equivariance.

## 4 Applications for Estimation and Tracking of Object Poses in 3D Space

### 4.1 Instance-level 6D Object Pose Estimation

Given an RGB-D image of a cluttered scene, instance-level 6D pose estimation is to estimate the 6D poses of known 3D objects with respect to the camera coordinate system. As introduced in Sec. 3.1, a 6D pose $\boldsymbol{g} \in \mathrm{SE}(3)$ can be decomposed into a 3D rotation $\boldsymbol{r} \in \mathrm{SO}(3)$ and a 3D translation $\boldsymbol{t} \in \mathbb{R}^3$, which makes sparse steerable convolutional network well suited for this task, due to: i) SS-Convs extract strong SE(3)-equivariant features to decode a precise 6D pose; ii) the steerability of feature maps helps to enable a second stage of pose refinement. Therefore, we propose an efficient general pipeline based on SS-Convs for 6D pose estimation, as depicted in Fig. 2.

Specifically, we firstly segment out the objects of interest via an off-the-shelf model of instance segmentation, assigning each object with an RGB segment and a cropped point cloud; then each 3D object is voxelized and represented by a sparse tensor $(\boldsymbol{H}_0, \boldsymbol{F}_0)$, where each feature in $\boldsymbol{F}_0$ is a $4-$dimensional vector, containing RGB values and a constant "1". For the input tensor, we set the site active if the quantified grid centered at this site encloses any points, and average point features of those enclosed by a same grid. $(\boldsymbol{H}_0, \boldsymbol{F}_0)$ is then fed into our pipeline in Fig. 2, where the pose estimation could be achieved in the following two stages.

In the first stage, we construct an efficient SS-Conv-based backbone, which extracts hierarchical SE(3)-equivariant feature maps, represented in the form of sparse tensors $\{(\boldsymbol{H}_n, \boldsymbol{F}_n)\}$. Those feature tensors are used for interpolation of multi-level point-wise features by using a Tensor-to-Point module, proposed in [10], transforming features of discretized grid sites to those of real-world point coordinates. Each point feature is fed into two separate MLPs, regressing a point offset and a rotation, respectively; the addition of the point coordinate and its offset generates a translation. The initially predicted pose $(\boldsymbol{r}_1, \boldsymbol{t}_1)$ of this stage is obtained by averaging point-wise predictions.

In the second stage, we refine the pose $(\boldsymbol{r}_1, \boldsymbol{t}_1)$ by learning a residual pose $(\boldsymbol{r}_2, \boldsymbol{t}_2)$, wherein a Feature-Steering module is designed, generating transformed features $\{(\boldsymbol{H}_n', \boldsymbol{F}_n')\}$ by efficiently steering hierarchical backbone features $\{(\boldsymbol{H}_n, \boldsymbol{F}_n)\}$ individually with $(\boldsymbol{r}_1, \boldsymbol{t}_1)$. Again we interpolate point-wise features from $\{(\boldsymbol{H}_n', \boldsymbol{F}_n')\}$, and average point-wise predictions to obtain $(\boldsymbol{r}_2, \boldsymbol{t}_2)$. Finally, the predicted 6D pose is updated as $(\boldsymbol{r}_1\boldsymbol{r}_2, \boldsymbol{t}_1 + \boldsymbol{r}_1\boldsymbol{t}_2)$. In addition, owing to the novel Feature-Steering modules, this stage can be iteratively repeated, generating finer and finer poses.

### 4.1.1 The Feature-Steering Module

Feature-Steering module in the pipeline is to transform $(\boldsymbol{H}_n, \boldsymbol{F}_n)$ of the backbone to $(\boldsymbol{H}_n', \boldsymbol{F}_n')$, where a rigid transformation of $\boldsymbol{H}_n$ with $(\boldsymbol{r}, \boldsymbol{t})$ and a rotation of $\boldsymbol{F}_n$ with $\rho(\boldsymbol{r})$ are included. Specifically, for $\boldsymbol{F}_n$, we compute $\rho(\boldsymbol{r})$ as defined in (5) and rotate $\boldsymbol{F}_n$ by matrix multiplication; for $\boldsymbol{H}_n$, we convert the sites in it to the real-world point coordinates, which are then applied to a rigid transformation of $(\boldsymbol{r}, \boldsymbol{t})$ and re-voxelized as grid sites. The same new sites are merged to a unique one, while their features are averaged. We also use two another SS-Convs, each followed by steerable normalization and activation, to enrich the new features and generate the final steered $(\boldsymbol{H}_n', \boldsymbol{F}_n')$.

### 4.2 Category-level 6D Object Pose and Size Estimation

Category-level 6D pose and size estimation is formally introduced in [24]. This is a more challenging task, which aims to estimate categorical 6D poses of unknown objects, and also the 3D object sizes. To tackle this problem, we use a similar network as that in Fig. 2, and make some adaptive modifications: i) for each stage in Fig. 2, we add another two separate MLPs for point-wise predictions of 3D sizes and point coordinates in the canonical space, respectively; ii) in each Feature-Steering module, the real-world coordinates of all 3D objects are also scaled by their predicted 3D sizes to be enclosed within a unit cube, for estimating more precise poses.

### 4.3 Category-level 6D Object Pose Tracking

Motivated by the above task of categorical pose estimation, category-level 6D pose tracking is also proposed to estimate the small change of 6D poses in two adjacent RGB-D frames of an image sequence [22]. Due to the available pose of the previous frame, the target object can be roughly located in the current frame, avoiding the procedures of object detection or instance segmentation in images. However, without a precise mask, the estimation of small pose change from noisy 3D data is a big challenge for deep networks. Our sparse steerable convolutional network also surprisingly performs well in such noisy data, even though we only conduct one-stage pose estimation that achieves real-time tracking. For more details, one may refer to the supplementary material.

## 5 Experiments

**Datasets** We conduct experiments on the benchmark LineMOD dataset [11] for instance-level 6D pose estimation, which consists of 13 different objects. For both category-level 6D pose estimation and tracking, we experiment on REAL275 dataset [24], which is a more challenging real-world

Table 1: Quantitative comparisons of Plain12 based on different convolutions on the LineMOD dataset [11].

| Conv | ADD(S) ↑ | FPS ↑ | #Param ↓ |
|------|----------|-------|----------|
| Dense-Conv | 46.5 | 224 | 26.2 M |
| SP-Conv | 62.8 | 486 | 26.2 M |
| ST-Conv | 92.8 | 148 | 3.6 M |
| SS-Conv | 93.5 | 404 | 3.6 M |

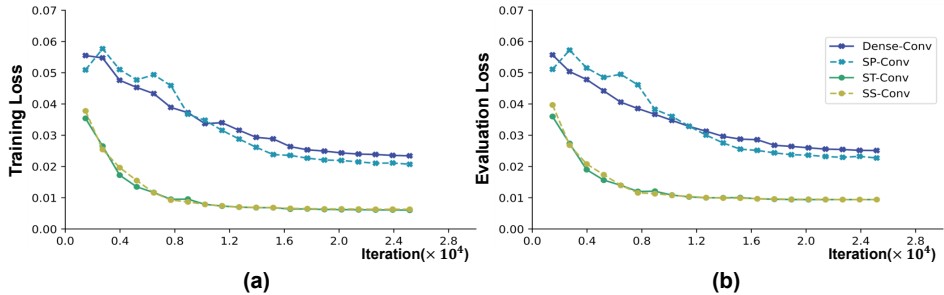

Figure 3: Training of Plain12 based on different convolutions on the LineMOD dataset [11].

dataset with $4,300$ training images and $2,750$ testing ones, containing object instances of 6 categories. Following [24, 22], we augment the training data of REAL275 with synthetic RGB-D images.

**Evaluation Metrics** For instance-level task, we follow [23] and evaluate the results of LineMOD dataset on ADD(S) metric. For the category-level tasks, we report the mean Average Precision (mAP) of intersection over union (IoU) and $n°m$ cm, following [24]; mean rotation error ($\boldsymbol{r}_{err}$) in degrees and mean translation error ($\boldsymbol{t}_{err}$) in centimeters are also reported for pose tracking. Additionally, we compare the numbers of parameters (#Param) and the running speeds (FPS) for different models. Testing is conducted on a server with a GeForce RTX 2080ti GPU for a batch size of 32, and FPS is computed by averaging the time cost of forward propagation on the whole dataset.

## 5.1 Comparisons with Different 3D Convolutions

We firstly conduct experiments to compare our proposed SS-Conv with other kinds of 3D convolutions, including conventional 3D convolution (Dense-Conv), sparse convolution (SP-Conv) [9], and steerable convolution (ST-Conv) [25], on the LineMOD dataset for instance-level 6D pose estimation. Among those convolutions, SP-Conv improves the speed of Dense-Conv by considering data sparsity and turns out to be efficient in some tasks of 3D semantic analysis(*e.g.*, 3D object detection), while ST-Conv constructs rotation-steerable kernels and then realizes the convolution based on Dense-Conv.

To meet various computational demands of different convolutions, those experiments are conducted on a light plain architecture, termed as **Plain12**, in the same experimental settings, for a fair comparison. The architecture consists of 12 convolutional layers, of which the kernel sizes are all set as $3 \times 3 \times 3$; for SS-Convs and ST-Convs, we set the superparameters of the radial function $\varphi^m$ in Eq. (8) as $\{m\} = \{0, 1\}$ and $\epsilon = 0.6$. We use ADAM to train the networks for a total of $30,000$ iterations, with an initial learning rate of $0.01$, which is halved every $1,500$ iterations. We voxelize the input segmented objects into $64 \times 64 \times 64$ dense/sparse grids, and set the training batch size as 16.

Quantitative results of different convolutions are listed in Table 1, which confirms the advantages of our SS-Conv in both accuracy and efficiency. In terms of accuracy, SS-Conv achieves comparable results on ADD(S) metric as ST-Conv does, which significantly outperforms those of Dense-Conv and SP-Conv, indicating the importance of SE(3)-equivariant feature learning on pose estimation;with preservation of relative poses of features layer-by-layer, the property of SE(3)-equivariance makes feature learning capture more information of object poses. We also visualize the behaviors of the four convolutions in the process of training in Fig. 3, where the learning based on SS-Conv/ST-Conv converges better and faster than that of Dense-Conv/SP-Conv.

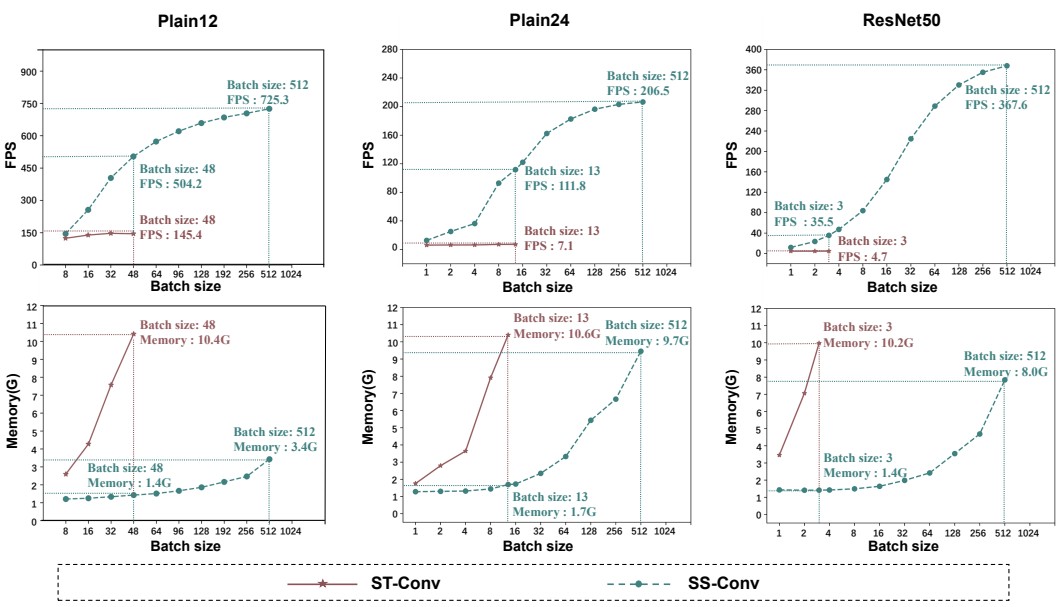

Figure 4: Plottings of FPS and memory consumption versus different batch sizes for different networks based on ST-Conv/SS-Conv. Experiments are conducted on LineMOD dataset[11].

Table 2: Quantitative comparisons of different methods on the LineMOD dataset [11] for instance-level 6D object pose estimation. The evaluation metric is ADD(S).

|         | Implicit[18] +ICP | SSD6D[13] +ICP | PointFusion [27] | DenseFusion [23] | DenseFusion (Iterative)[23] | G2L[3] | Ours w/o second stage | Ours |
|---------|------|------|------|------|------|------|------|------|
| ape     | 20.6 | 65   | 70.4 | 79.5 | 92.3 | 96.8 | 92.9 | **97.4** |
| bench.  | 64.3 | 80   | 80.7 | 84.2 | 93.2 | 96.1 | 97.4 | **99.3** |
| camera  | 63.2 | 78   | 60.8 | 76.5 | 94.4 | 98.2 | 97.7 | **99.5** |
| can     | 76.1 | 86   | 61.1 | 86.6 | 93.1 | 98.0 | 96.1 | **99.6** |
| cat     | 72.0 | 70   | 79.1 | 88.8 | 96.5 | 99.2 | 98.8 | **99.8** |
| driller | 41.6 | 73   | 47.3 | 77.7 | 87.0 | **99.8** | 98.7 | 99.6 |
| duck    | 32.4 | 66   | 63.0 | 76.3 | 92.3 | 97.7 | 91.1 | **97.8** |
| **egg.** | 98.6 | 100  | 99.9 | 99.9 | 99.8 | **100.0** | 100.0 | 99.9 |
| **glue** | 96.4 | 100  | 99.3 | 99.4 | **100.0** | 100.0 | 98.6 | 99.6 |
| hole.   | 49.9 | 49   | 71.8 | 79.0 | 92.1 | 99.0 | 96.3 | **99.4** |
| iron    | 63.1 | 78   | 83.2 | 92.1 | 97.0 | **99.3** | 98.7 | 99.2 |
| lamp    | 91.7 | 73   | 62.3 | 92.3 | 95.3 | 99.5 | 99.5 | **99.7** |
| phone   | 71.0 | 79   | 78.8 | 88.0 | 92.8 | **98.9** | 97.5 | 98.2 |
| MEAN    | 64.7 | 79   | 73.7 | 86.2 | 94.3 | 98.7 | 97.2 | **99.2** |

In terms of efficiency, our sparse steerable convolutional networks are more efficient and flexible for complex systems, *e.g.*, for Plain12, SS-Conv brings about 2.7× speedup w.r.t. ST-Conv (404 FPS versus 148 FPS) with a batch size of 32, as listed in Table 1. More results of FPS with improved sizes of data batches are given in Fig. 4, where ST-Conv can be only run at the extreme batch size of 48 on the GPU with 12G memory, while SS-Conv costs much less memory and can thus support a batch size as large as 512; running with larger batch sizes further improves the efficiency of our proposed SS-Conv (FPS goes to 725 when running with the batch size of 512 on Plain12). We also compare the efficiency of ST-Conv and SS-Conv on two other deeper networks (dubbed **Plain24** and **ResNet50**, respectively); as shown in Fig. 4, SS-Conv consistently improves FPS over ST-Conv on these two architectures, with less GPU memory consumption.

## 5.2 Comparisons with Existing Methods

**Instance-level 6D Object Pose Estimation** For the instance-level task, we compare the results of our SS-Conv-based pipeline with existing methods on LineMOD dataset [11]. Quantitative results

Table 3: Quantitative comparisons of different methods on REAL275 dataset [24] for category-level 6D object pose and size estimation.

| Method | mAP | | | | | |
|---|---|---|---|---|---|---|
| | $IoU_{50}$ | $IoU_{75}$ | $5°2cm$ | $5°5cm$ | $10°2cm$ | $10°5cm$ |
| NOCS [24] | 78.0 | 30.1 | 7.2 | 10.0 | 13.8 | 25.2 |
| SPD [21] | 77.3 | 53.2 | 19.3 | 21.4 | 43.2 | 54.1 |
| CASS [2] | 77.7 | – | – | 23.5 | – | 58.0 |
| FS-Net [4] | **92.2** | 63.5 | – | 28.2 | – | 60.8 |
| DualPoseNet [15] | 79.8 | 62.2 | 29.3 | 35.9 | 50.0 | **66.8** |
| Ours w/o second stage | 79.5 | 58.7 | 19.2 | 25.2 | 35.1 | 49.9 |
| Ours | 79.8 | **65.6** | **36.6** | **43.4** | **52.6** | 63.5 |

Table 4: Quantitative comparisons of different methods on REAL275 dataset [24] for category-level 6D object pose tracking.

| Method | Metric | bottle | bow | camera | can | laptop | mug | MEAN |
|---|---|---|---|---|---|---|---|---|
| 6-PACK [22] | $5°5cm \uparrow$ | 24.5 | 55.0 | 10.1 | 22.6 | 63.5 | 24.1 | 33.3 |
| | $IoU_{25} \uparrow$ | 91.1 | 100.0 | 87.6 | 92.6 | 98.1 | 95.2 | 94.1 |
| | $r_{err} \downarrow$ | 15.6 | 5.2 | 35.7 | 13.9 | 4.7 | 21.3 | 16.1 |
| | $t_{err} \downarrow$ | 4.0 | 1.7 | 5.6 | 4.8 | 2.5 | 2.3 | 3.5 |
| Ours | $5°5cm \uparrow$ | 70.3 | 60.6 | 10.6 | 49.9 | 87.7 | 47.9 | 54.5 |
| | $IoU_{25} \uparrow$ | 93.5 | 99.9 | 99.9 | 99.8 | 99.8 | 99.9 | 98.8 |
| | $r_{err} \downarrow$ | 3.7 | 4.6 | 9.8 | 4.6 | 3.0 | 5.6 | 5.2 |
| | $t_{err} \downarrow$ | 1.9 | 1.2 | 2.0 | 2.7 | 2.4 | 1.1 | 1.9 |

are shown in Table 2, where our two-stage pipeline outperforms all the existing methods and achieves a new state-of-the-art result of $99.2\%$ on mean ADD(S) metric. We can also observe that the second stage of pose refinement with Feature-Steering modules in our pipeline indeed improves the predictions in the first stage, benefitting from the steerability of the feature spaces in SS-Convs.

**Category-level 6D Object Pose and Size Estimation** We conduct experiments on REAL275 [24] for the more challenging category-level task. Quantitative results in Table 3 confirm the advantage of our pipeline in the high-precision regime, especially on the precise metric of $5°5cm$, where we improve the state-of-the-art result in [15] from $35.9\%$ to $43.4\%$. The second stage of pose refinement also plays an important role in this task, achieving remarkable improvements over the first stage.

**Category-level 6D Object Pose Tracking** We compare the results of our one-stage tracking pipeline with the baseline of 6-PACK [22] on REAL275 [24]. In 6-PACK, the relative pose between two frames is computed based on predicted keypoint pairs inefficiently, while our pipeline regresses the pose in a direct way. The results in Table 4 show that our pipeline outperforms 6-PACK on all the evaluation metrics, demonstrating the ability of SS-Conv-based network for fine-grained pose estimation in noisy input data.

*More implementation details and qualitative results are shown in the supplementary material.*

## Broader Impact

The studied problems of object pose estimation and tracking in 3D space are very important to many real-world applications, including augmented reality, robotic grasping, and autonomous driving. By precisely predicting object poses in the 3D space, virtual contents could be seamlessly embedded in real environments, creating fascinating personal experience; on the contrary, less precise predictions may cause property loss and even life threat, especially in autonomous driving. The contributed solution based on SS-Conv would improve the overall level of safety.

## Acknowledgments and Funding Disclosure

This work was partially supported by the Guangdong R&D key project of China (No.: 2019B010155001), the National Natural Science Foundation of China (No.: 61771201), and the Program for Guangdong Introducing Innovative and Entrepreneurial Teams (No.: 2017ZT07X183).

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
