# OpenReview forum: "Sparse Steerable Convolutions: An Efficient Learning of SE(3)-Equivariant Features for Estimation and Tracking of Object Poses in 3D Space"
_NeurIPS.cc/2021/Conference — NeurIPS 2021 Poster_

### Official Review · Reviewer_hf1K · 2021-07-12

**Rating:** 7
**Confidence:** 2

**Summary:**

This paper extends the concepts of sparse manifold convolution to the steerable convolution. In particular, a sparse steerable convolution operation is proposed to reduce the redundant computation in volumetric data, while maintaining the SE(3) equivariance. Extensive experiments conducted on three tasks (i.e., instance-level 6D pose estimation, category-level 6D pose and size estimation, and category-level 6D pose tracking) demonstrated the efficiency and effectiveness of the proposed SS-Conv.


**Main Review:**

Overall, the idea of this paper is straightforward and feasible. It is desirable to extend the sparse convolution to steerable convolution, especially in light of the expensive computational cost.  The reported experimental results also demonstrate better efficiency of the proposed SSConv operation.

However, the real technical contribution of this paper is somewhat limited, since the sparse convolution has been well-developed and achieving remarkable performance in other fields such as semantic segmentation. In addition, several well-optimized libraries such as SparseConvNet and Minkowski engine further simplified the implementation.

Experiments:

-Please increase the readability of Figure 3, the fonts are too small to be recognized.

-Learning generalized local descriptors from 3D point clouds also requires rotation and translation invariance. I am wondering if the proposed SSConv can be integrated into the FCGF [1] framework to achieve better registration performance.

[1] Fully convolutional geometric features. In ICCV, 2019.

-Please also report the FLOPS of different implementations in Table 1.

-In Table 2, the performance of the proposed approach is inferior to G2L [3] when the second stage is not used, what is the possible reason? what is the impact of the second stage on the proposed framework? What is the extra computational cost introduced by the second stage?


Minor comments:

Line 82, the reference is missing.

**Time Spent Reviewing:**

4 hours

---

> ### Author Response · Authors · 2021-08-10
> **Authors' Response**
>
> We appreciate the reviewer’s positive comments on the paper, and will address the reviewer’s concerns individually as follows.
>
> ## About novelty
>
> In terms of technical contributions, we do agree that existing works inspires the present one, and designs of SS-Conv of benefit from sparse convolution libraries. However, we have to emphasize that a trivial combination of ST-Conv and SP-Conv is not enough to develop a strided SS-Conv. As pointed out in [24], smoothing feature maps is very critical to ST-Conv in the process of down-sampling, and thus, it is implemented as a combination of an ST-Conv without spatial reduction and a strided convolution with a Gaussian kernel, instead of a single strided convolution.  Technically, we develop a new and efficient strided SS-Conv with smooth feature maps in our implementation. Specifically, we firstly apply a convolution with a Gaussian kernel on the kernel of SS-Conv, which is enlarged without increasing learnable parameters (e.g., the kernel of SS-Conv is $3\times3\times3\times32\times32$, and the Gaussian kernel is $3\times3\times3$ with stride 1 and padding 1, then the kernel is enlarged as $5\times5\times5\times32\times32$); with the enlarged kernel, strided SS-Conv can be completed via ALGORITHM 1 in Sec. 3.2.3. Compared to the way in [24], our implementation is more efficient since convolution on kernel costs much less than that on the feature map. We also note that this way is not applicable to ST-Conv, because enlarging kernels in dense convolution results in cubic computational cost. We will present more details of the strided SS-Conv in the paper.
>
> In addition, we propose a general pipeline in the paper (cf. Fig. 2) for pose-related semantic tasks, and technical innovations are needed as well in order to fully exploit the potential of SS-Conv. For example, based on the steerability of the learned features, we design a Feature-Steering module to enable a second stage of iterative pose refinement, such that it becomes unnecessary to train another residual pose estimation network as did in [22]; the second stage of refinement not only gives us finer pose predictions by learning residual poses, but also enhances the SE(3)-equivariant feature learning by providing auxiliary supervision. We also argue that compared to ST-Conv and SP-Conv, SS-Conv is an optimal choice for such a pipeline. Specifically, for ST-Conv, the use of all grid features in a dense voxel makes it inefficient to interpolate point-wise features via Tensor-to-Point Module, and even contaminates them due to the involvement of grid features located at blank areas; for SP-Conv, due to the lack of steerability, an independent residual pose estimation network becomes necessary in order to learn features from raw 3D data transformed by poses of the first stage.
>
> In the paper, we also confirm the efficacy of SS-Conv on three pose-related semantic applications, outperforming existing methods under almost all the metrics. As pointed out by Reviewer aue9, these applications alone could be a paper in a mainstream vision conference.
>
> &nbsp;
>
> ## About experiments
>
> Next, we address the reviewer’s concerns on experiments individually.
>
> __Q1: Please increase the readability of Figure 3, the fonts are too small to be recognized.__
>
> __Reply:__ Thanks for the suggestion and we will increase font sizes in Fig. 3 to improve the readability.
>
> __Q2: Learning generalized local descriptors from 3D point clouds also requires rotation and translation invariance. I am wondering if the proposed SSConv can be integrated into the FCGF [1] framework to achieve better registration performance.__
>
> __Reply:__ We appreciate the comment on integrating SS-Conv into the FCGF [1] framework for better registration. We agree with the reviewer and think that this is technically possible. More specifically, SS-Conv can be integrated into FCGF by simply replacing the original sparse convolutions; then the rotation-equivariant output features of the backbone could be transformed into rotation-invariant ones with order l=0 via another $1\times1\times1$ SS-Conv. By doing so, information from inputs, including shapes and orientations, is kept in the rotation-equivariant features along hierarchy as much as possible; instead, the original FCGF may cause information loss in the process of local invariant feature learning from the very beginning. Thus, FCGF with SS-Conv has the potential to improve the performance. We will try this idea in future research.
>
> [1] Fully convolutional geometric features. In ICCV, 2019.
>
> __Q3: Please also report the FLOPS of different implementations in Table 1.__
>
> __Reply:__ Thanks and we will include the FLOPS of different methods in Table 1.
>
> __Q4: In Table 2, the performance of the proposed approach is inferior to G2L [3] when the second stage is not used, what is the possible reason? what is the impact of the second stage on the proposed framework? What is the extra computational cost introduced by the second stage?__
>
> __Reply:__ For the second stage of our proposed method for instance-level 6D pose estimation, we first note that it is the key design of feature-steering module that enables the second stage (and even more stages) of iterative pose refinement. With the enabled pose refinement, our method makes it easier to give out finer pose predictions by learning residual ones, whose learning space is marginally narrowed from the whole SE(3) space; meanwhile, since both stages share the same backbone, the learning of SE(3)-equivariant feature is also enhanced with auxiliary supervision of the second stage. When the second stage is not used, the reviewer points out that our method is inferior to G2L [3]; we argue that in fact, G2L implements the process of rotation estimation in 2 steps as we do, with an initial rotation and a residual one simultaneously learned in its framework. Besides, G2L trains an individual model for each object instance in their code implementation, while we only use a single model for all objects. We finally note that the computational cost in the second stage is light, including feature-steering modules, tensor-to-point module, and MLPs for residual pose estimation, as illustrated in Fig. 2. For pose refinement, we avoid using another backbone to extract features of inputs transformed by the poses of the first stage (e.g., as DenseFusion [22] does), thanks to the use of feature-steering modules; compared to the backbone, especially to a very deep one, the extra computation of our feature-steering modules is much less cost.
>
> __Q5: Line 82, the reference is missing.__
>
> __Reply:__ Line 82 should be corrected as “Wang et. al [21] propose …”. We will include the reference in the paper.

---

> > ### Comment · Reviewer_hf1K · 2021-08-28
> > **post-rebuttal comments**
> >
> > The rebuttal has solved most of my concerns. After carefully reading all comments from other reviewers, I would upgrade the rating from 6 to 7.

---

### Official Review · Reviewer_zbbB · 2021-07-15

**Rating:** 5
**Confidence:** 3

**Summary:**

This work proposes a reformulation of Steerable Convolutions (ST-Conv) , dubbed Sparse Steerable Convolutions (SS-Conv),  that allows them to run on sparse voxel grid while still preserving SO3 equivariance.

Specifically, accelerations is achieved in two ways: 1) convolutions are run only on activated voxels, 2) in the receptive field of activated sites, only active input features are convolved.
To further increase efficiency and alleviate the "dilation" problem of sparse convolutions, authors also propose a modification to SS-Convs, dubbed Submanifold SS-Convs, where output voxel state is kept consistent with the input one, making it possible to construct a deep but efficient network. In practice Manifold SS-Convs are alternated to SS-Convs to guarantee high performance and efficeincy.

The proposed method is first compared to ST-Convs, Dense Convs, and Sparse convs on the LineMOD dataset, where it demonstrates similar performance to ST-Convs while being 2x faster. Furthermore, it is validated through the following 3D semantical analysis tasks: instance-level 6D pose estimation, category-level 6D pose and size estimation, and category level 6D pose tracking.


**Limitations And Societal Impact:**

Yes.

**Main Review:**

Strengths:
- Sparse implementation of ST-Convs could be beneficial for community working with sparse 3D data, as allows for efficiency that is on par with standard 3D convolutions
- Proposed application-specific architectures exploit Steerable convolutions to achieve state-of-the-art performance and demonstrate the effectiveness on steerable convolutions on tasks that had not been tackled by this architecture before

Weaknesses:
- Novelty: a more efficient implementation of an existing method is not enough novelty for NeurIPS, especially given that for the presented experiments SS-Convs only result in a 2x speedup wrt ST-Convs, and authors have to actually use a single-stage architecture to reach real-time performance on 6D object pose tracking.

Questions for authors:
- In Table 1, SS-Convs outperforms ST-Convs. How do you justify this, given that SS-Convs is a sparse implementation of ST-Convs? Are there any other differences between the two considered architectures, or is it due to the use of Submanifold SS-Convs whitin the architecture?  If so, it should be specified.

- Are Submanifold convolutions. novel? From the paper, it is hard to say if the if the idea of using Submanifold SS-Convs is novel, or if it has already been propose for sparse 3D convolutions and adapted to this framework.



**Time Spent Reviewing:**

4

---

> ### Author Response · Authors · 2021-08-10
> **Authors' Response**
>
> We appreciate the reviewer’s comments on the strengths of our proposed Sparse Steerable Convolutions (SS-Conv). We further explain that the efficiency of SS-Conv improves with the increased sizes of data batches during both training and inference (cf. Fig. 2 in the supplementary material); for example, for a plain backbone consisting of standard conv layers (dubbed Plain24 in Fig. 2), SS-Conv brings about 16x speedup w.r.t. ST-Conv (111.8 FPS versus 7.1 FPS) at the batch size of 13; more importantly, SS-Conv costs much less memory and can thus support a batch size as large as 512, while ST-Conv can be only run at the extreme batch size of 13 on the GPU with 12G memory; running with larger batch sizes further improves the efficiency of our proposed SS-Conv (as shown in Fig. 2, FPS goes to 206.5 when running with the batch size of 512).
>
> For technical contributions of SS-Conv, it is indeed based on STeerable convolution (ST-Conv) [24] and submanifold SParse convolution (SP-Conv) [9]; however, we have to emphasize that a trivial combination of ST-Conv and SP-Conv is not enough to develop a strided SS-Conv. As pointed out in [24], smoothing feature maps is very critical to ST-Conv in the process of down-sampling, and thus, it is implemented as a combination of an ST-Conv without spatial reduction and a strided convolution with a Gaussian kernel, instead of a single strided convolution.  Technically, we develop a new and efficient strided SS-Conv with smooth feature maps in our implementation. Specifically, we firstly apply a convolution with a Gaussian kernel on the kernel of SS-Conv, which is enlarged without increasing learnable parameters (e.g., the kernel of SS-Conv is $3\times3\times3\times32\times32$, and the Gaussian kernel is $3\times3\times3$ with stride 1 and padding 1, then the kernel is enlarged as $5\times5\times5\times32\times32$); with the enlarged kernel, strided SS-Conv can be completed via ALGORITHM 1 in Sec. 3.2.3. Compared to the way in [24], our implementation is more efficient since convolution on kernel costs much less than that on the feature map. We also note that this way is not applicable to ST-Conv, because enlarging kernels in dense convolution results in cubic computational cost. We will present more details of the strided SS-Conv in the paper.
>
> In addition, we propose a general pipeline in the paper (cf. Fig. 2) for pose-related semantic tasks, and technical innovations are needed as well in order to fully exploit the potential of SS-Conv. For example, based on the steerability of the learned features, we design a Feature-Steering module to enable a second stage of iterative pose refinement, such that it becomes unnecessary to train another residual pose estimation network as did in [22]; the second stage of refinement not only gives us finer pose predictions by learning residual poses, but also enhances the SE(3)-equivariant feature learning by providing auxiliary supervision. We also argue that compared to ST-Conv and SP-Conv, SS-Conv is an optimal choice for such a pipeline. Specifically, for ST-Conv, the use of all grid features in a dense voxel makes it inefficient to interpolate point-wise features via Tensor-to-Point Module, and even contaminates them due to the involvement of grid features located at blank areas; for SP-Conv, due to the lack of steerability, an independent residual pose estimation network becomes necessary in order to learn features from raw 3D data transformed by poses of the first stage.
>
> Table 1 in the paper shows that SS-Conv performs slightly better than ST-Conv does (93.5 versus 92.8 under the metric of ADD(S)). One may be interesting on why this happens, if assuming that SS-Conv is only a sparse implementation of ST-Conv. In fact, we think the improvement comes from the use of Submanifold SS-Conv (cf. L149-L154 in Sec. 3.2.2), which restricts the quick feature dilations and better keeps the object contours on the feature maps (cf. Fig. 1 in the paper). Thanks for the suggestion and we will specify these factors more clearly in the paper that explain why SS-Conv brings performance improvement over ST-Conv.

---

> > ### Comment · Reviewer_zbbB · 2021-09-10
> > **Post-rebuttal" novelty issues?**
> >
> > After having read all reviews and author response, I am getting more and more convinced that this work perhaps doesn't reach the novelty bar for NeurIPS. Because of that, I have upgraded my recommendation.

---

### Official Review · Reviewer_BHAN · 2021-07-16

**Rating:** 6
**Confidence:** 4

**Summary:**

This paper extends steerable convolution by taking advantage of the sparsity of 3D data, such that the speed in both training and inference is significantly faster than the original one [24]. The method itself is motivated by SP-Conv [9], e.g. the site state definition and “submanifold dilation”. In evaluation, it is demonstrated that the method outperforms or is comparable to the state-of-the-arts in 3D object pose estimation, tracking and size estimation.

**Limitations And Societal Impact:**

Yes.

**Main Review:**

The paper is well-written, as it provides a gentle review of the background works which this one is built upon, and most of its discussion about related works and motivations is adequate and precise.  I’m also fully convinced by the practical impact of this work -- the noticeable speed up of steerable convolution is certainly something the community desires.

However, as a reviewer, I do concern the “novelty” of this work, as it reads to me, is "just" applying the engineering details proposed by  [9], e.g. “submanifold conv” to steerable conv [24].  I fail to see the technical novelty which is actually owned by this paper. I’d like to hear from other reviewers regarding this, and the author is of course welcomed to point out their contribution if I unfortunately missed.

Other minors regarding to quantitative comparison: (1) It would be nice if the authors also compare the running speed in table 2-4, or at least comment on their relative speeds. This potentially could make the readers appreciate more about the practical impact of this work.

(2) In table3, the proposed method leads in almost every metric, except IoU50, which shows a big gap when compared to  FS-Net[4]’s 92.2% accuracy. It seems that the proposed method has more catastrophic failures. Could the author show / comment on the type / reason of failures.


**Time Spent Reviewing:**

4

---

> ### Author Response · Authors · 2021-08-10
> **Authors' Response**
>
> We highly appreciate the reviewer’s comments and confirming on the practical impact of this work, and our responses to the reviewer’s concerns are made individually as follows.
>
> ## About novelty
>
> The proposed Sparse Steerable Convolution (SS-Conv) is indeed based on STeerable convolution (ST-Conv) [24] and submanifold SParse convolution (SP-Conv) [9]; We thank the reviewer for agreeing with the usefulness of our proposed Sparse Steerable Convolution (SS-Conv) for tasks of 3D semantic analysis. It is true that SS-Conv is based on previous works of STeerable convolution (ST-Conv) [24] and submanifold SParse convolution (SP-Conv) [9]; however, we have to emphasize that a trivial combination of ST-Conv and SP-Conv is not enough to develop a strided SS-Conv. As pointed out in [24], smoothing feature maps is very critical to ST-Conv in the process of down-sampling, and thus, it is implemented as a combination of an ST-Conv without spatial reduction and a strided convolution with a Gaussian kernel, instead of a single strided convolution.  Technically, we develop a new and efficient strided SS-Conv with smooth feature maps in our implementation. Specifically, we firstly apply a convolution with a Gaussian kernel on the kernel of SS-Conv, which is enlarged without increasing learnable parameters (e.g., the kernel of SS-Conv is $3\times3\times3\times32\times32$, and the Gaussian kernel is $3\times3\times3$ with stride 1 and padding 1, then the kernel is enlarged as $5\times5\times5\times32\times32$); with the enlarged kernel, strided SS-Conv can be completed via ALGORITHM 1 in Sec. 3.2.3. Compared to the way in [24], our implementation is more efficient since convolution on kernel costs much less than that on the feature map. We also note that this way is not applicable to ST-Conv, because enlarging kernels in dense convolution results in cubic computational cost. We will present more details of the strided SS-Conv in the paper.
>
> In addition, we propose a general pipeline in the paper (cf. Fig. 2) for pose-related semantic tasks, and technical innovations are needed as well in order to fully exploit the potential of SS-Conv. For example, based on the steerability of the learned features, we design a Feature-Steering module to enable a second stage of iterative pose refinement, such that it becomes unnecessary to train another residual pose estimation network as did in [22]; the second stage of refinement not only gives us finer pose predictions by learning residual poses, but also enhances the SE(3)-equivariant feature learning by providing auxiliary supervision. We also argue that compared to ST-Conv and SP-Conv, SS-Conv is an optimal choice for such a pipeline. Specifically, for ST-Conv, the use of all grid features in a dense voxel makes it inefficient to interpolate point-wise features via Tensor-to-Point Module, and even contaminates them due to the involvement of grid features located at blank areas; for SP-Conv, due to the lack of steerability, an independent residual pose estimation network becomes necessary in order to learn features from raw 3D data transformed by poses of the first stage.
>
> In the paper, we also confirm the efficacy of SS-Conv on three pose-related semantic applications, outperforming existing methods under almost all the metrics. As pointed out by Reviewer aue9, these applications alone could be a paper in a mainstream vision conference.
>
> &nbsp;
>
> ## About quantitative comparison
>
> __Q1: It would be nice if the authors also compare the running speed in table 2-4, or at least comment on their relative speeds.__
>
> __Reply:__ We thank the reviewer’s suggestions on additional quantitative comparisons. For comparison of running speed in Table 4, we compute FPS of both our proposed method and 6-PACK [21] on a server with GeForce RTX 2080ti GPU. As shown in the table below, our method achieves more exciting performance on speed than 6-PACK, which is also fast to enable real-time robot interaction; we note that we run the released code of 6-PACK to obtain its running speed, and find that it’s not as fast as claimed in the paper (~10 FPS), since it runs 27 times for each frame to get the reported results. For a fair comparison of running speed in Tables 2-3, we will also evaluate FPS of different methods on the same server accordingly.
>
> | &emsp;Method        |   &ensp;FPS  |
> |:-------------:|:-------------:|
> |6-PACK [21]    |         ~3.5  |
> |Ours           |         &ensp;~11.4 |
>
>
> __Q2: In table3, the proposed method leads in almost every metric, except IoU50, which shows a big gap when compared to FS-Net[4]’s 92.2% accuracy. Could the author show / comment on the type / reason of failures.__
>
> __Reply:__ Table 3 shows that our method performs worse than FS-Net [4] under the only metric of IoU50; unfortunately, the authors of [4] haven’t released all the estimated results or the trained models, so we could not qualitatively compare the results. We make some reasonable guesses as follows:
> 1) FS-net benefits from more precise object sizes by learning residual ones, while we directly learn the absolute ones. As we know, 6D object pose and size codetermine the values of IoU. For the coarse metric of IoU50, the precision of sizes makes a decision, since our method predicts more precise 6D poses (cf. the results on metrics of {n degree m cm}); for a stricter metric of IoU75, however, the precision of 6D poses plays a key role, and thus our method outperforms FS-Net.
> 2) We only use a single model to handle objects from all the 6 categories, while in FS-Net, individual models are trained for each category (Link: https://github.com/DC1991/FS_Net/issues/4).

---

> > ### Comment · Reviewer_BHAN · 2021-09-11
> > **post-rebuttal comments**
> >
> > Thanks to the authors response. After reading the rebuttal and others comments, I feel the novelty of the paper is less of an issue for me, thus I would keep the current rating.

---

### Official Review · Reviewer_viJM · 2021-07-18

**Rating:** 5
**Confidence:** 3

**Summary:**

This paper proposes sparse steerable convolution that efficiently learns SE(3)-equivariant features and demonstrates its usefulness in object pose estimation tasks. Essentially, the proposed method is a combination of 3D steerable cnn [24] and submanifold sparse convolutional networks [9]. The proposed method, being able to extract SE(3)-equivariant features, is proven to be useful in 6D pose estimation, outperforming previous state-of-the-art methods

**Limitations And Societal Impact:**

It would be helpful if the authors highlight failure cases of this work compared to a regular sparse convolution neural networks.

**Main Review:**

This work to my best understanding is a nice engineering effort to bring 3D steerable convolution to sparse voxels. The experiments show that the property of SE(3)-equivariance extracts better information of object poses while being more efficient than its dense counterpart.

However, I am concerned about the novelty of this work. To my best understanding, almost all of the theoretical basis of 3D steerable convolution came from its previous work [24] and submanifold sparse convolutional networks [9], and this work mostly reiterates previous literature. I am not sure what makes this novel, especially considering the venue of the conference. Although I agree that in engineering perspective, the proposed sparse steerable convolutions is efficient and reasonable, considering the venue of the conference, I am not sure if this work satisfies the novelty claim that it makes. If there is any unique problem being sparse, I believe it should better be highlighted. The comparison in the supplementary material only discusses engineering perspectives such as speed or memory consumption, which is the novelty of sparse convolution, not this work.


**Time Spent Reviewing:**

3hrs

---

> ### Author Response · Authors · 2021-08-10
> **Authors' Response**
>
> We thank the reviewer for agreeing with the usefulness of our proposed Sparse Steerable Convolution (SS-Conv) for tasks of 3D semantic analysis. It is true that SS-Conv is based on previous works of STeerable convolution (ST-Conv) [24] and submanifold SParse convolution (SP-Conv) [9]; however, we have to emphasize that a trivial combination of ST-Conv and SP-Conv is not enough to develop a strided SS-Conv. As pointed out in [24], smoothing feature maps is very critical to ST-Conv in the process of down-sampling, and thus, it is implemented as a combination of an ST-Conv without spatial reduction and a strided convolution with a Gaussian kernel, instead of a single strided convolution.  Technically, we develop a new and efficient strided SS-Conv with smooth feature maps in our implementation. Specifically, we firstly apply a convolution with a Gaussian kernel on the kernel of SS-Conv, which is enlarged without increasing learnable parameters (e.g., the kernel of SS-Conv is $3\times3\times3\times32\times32$, and the Gaussian kernel is $3\times3\times3$ with stride 1 and padding 1, then the kernel is enlarged as $5\times5\times5\times32\times32$); with the enlarged kernel, strided SS-Conv can be completed via ALGORITHM 1 in Sec. 3.2.3. Compared to the way in [24], our implementation is more efficient since convolution on kernel costs much less than that on the feature map. We also note that this way is not applicable to ST-Conv, because enlarging kernels in dense convolution results in cubic computational cost. We will present more details of the strided SS-Conv in the paper.
>
> In addition, we propose a general pipeline in the paper (cf. Fig. 2) for pose-related semantic tasks, and technical innovations are needed as well in order to fully exploit the potential of SS-Conv. For example, based on the steerability of the learned features, we design a Feature-Steering module to enable the second stage of iterative pose refinement, such that it becomes unnecessary to train another residual pose estimation network as did in [22]; the second stage of refinement not only gives us finer pose predictions by learning residual poses, but also enhances the SE(3)-equivariant feature learning by providing auxiliary supervision. We also argue that compared to ST-Conv and SP-Conv, SS-Conv is an optimal choice for such a pipeline. Specifically, for ST-Conv, the use of all grid features in a dense voxel makes it inefficient to interpolate point-wise features via Tensor-to-Point Module, and even contaminates them due to the involvement of grid features located at blank areas; for SP-Conv, due to the lack of steerability, an independent residual pose estimation network becomes necessary in order to learn features from raw 3D data transformed by poses of the first stage.
>
> In the paper, we also confirm the efficacy of SS-Conv on three pose-related semantic applications, outperforming existing methods under almost all the metrics. As pointed out by Reviewer aue9, these applications alone could be a paper in a mainstream vision conference.
>
> We note again that SE(3)-equivariant deep feature learning is essential for pose-related 3D semantic tasks; it is yet less explored until very recently, possibly due to the less convenient usage of existing techniques (e.g., the huge time- and memory-costs for steerable convolutions). We expect that SS-Conv proposed in the present paper would improve the convenience of learning SE(3)-equivariant features for pose-related 3D semantic tasks, and consequently would inspire new algorithms that perform even better on these tasks.
>
> We also appreciate the reviewer’s suggestion on showing the failure cases of our method when compared with SP-Conv based networks. In terms of quantitative measures, the network based on our proposed SS-Conv outperforms that on SP-Conv by a large margin of 30.7% on ADD(S) metric (93.5% v.s. 62.8%, cf. Table 1 in the submitted paper) on LineMOD dataset, under the same experimental conditions. Qualitatively, our method predicts highly precise poses for most target objects (cf. Fig. 5 in the supplementary material), while fails in a minority of cases with heavy occlusions or self-occlusions; however, for those failure cases, the deviations of our predictions against ground truths are still much smaller than those of SP-Conv based network.  We will supplement the qualitative comparisons of the failure cases predicted by those two methods to the paper accordingly.

---

> > ### Comment · Reviewer_viJM · 2021-09-01
> > **Update**
> >
> > Thank you so much for clarification.
> >
> > I see multiple reviewers' concerns about the novelty and the authors made a huge effort defending it. The authors find novelty in the following:
> >
> >  1. Efficient implementation using associativity of the filters: The authors in the paper claimed the novelty of the entire SS-Conv whereas in the rebuttal the authors emphasized a detail in the implementation. If the novelty of this work is in the enlarged kernel, this paper should be properly ablated and quantitatively studied it, and clearly stated that this specific part of implementation is what makes this paper novel. Without ablating it, the authors should not make claims like
> > > Compared to the way in [24], our implementation is more efficient since convolution on kernel costs much less than that on the feature map.
> >
> > because we are not sure how much of efficiency is gained in sparse convolution vs the way this paper implemented it. My guess without such experiments is that most of the efficiency is coming from the sparse operation itself, not the way this paper merged the kernels.
> >
> > 2. The pipeline: I would totally agree that this is a meaningful novelty if it were computer vision conference. And I could agree with
> > > As pointed out by Reviewer aue9, these applications alone could be a paper in a mainstream vision conference.
> >
> > this. I am just arguing the novelty of this work as a NeurIPS paper, since NeurIPS is not a vision conference.
> >
> > I would defend this work for accept if it were computer vision conference. I would be happy to see this paper accepted, but my concerns are still not well addressed, and I see no reason to update my rating.

---

### Official Review · Reviewer_hnKB · 2021-07-18

**Rating:** 8
**Confidence:** 4

**Summary:**

This paper discusses the development of a novel method called Sparse Steerable Convolution for the estimation of an object’s 3D pose. As the pose estimation is usually done using SE(3)-equivariant deep feature learning, the proposed method provides more efficient computation using sparse tensors. The paper presents carefully the theoretical background of the method and experiments it via using two different benchmark image datasets for estimating the 6D object pose.

**Limitations And Societal Impact:**

Yes

**Main Review:**

The paper is well written, scientifically sound and provides enough details about the method. The method is novel and provides means for efficient and still accurate object pose estimation. The paper gives credit for previous work. Novelty and contribution are clearly stated. The theoretical premises are carefully explained. The method is tested and analyzed using two very different benchmark image datasets and the results are compared to multiple state-of-the-art object pose estimation methods.

**Time Spent Reviewing:**

3

---

> ### Author Response · Authors · 2021-08-10
> **Authors' Response**
>
> We highly appreciate the reviewer’s comments and want to thank the reviewer for summarizing key contributions of the submission.

---

### Official Review · Reviewer_aue9 · 2021-07-22

**Rating:** 9
**Confidence:** 5

**Summary:**

This paper builds upon Weiler's [24] steerable filters on volumetric data
and creates a new equivariance by design by sparsifying group convolutions.
Authors are inspired by Graham's [9] sparsification of convolution that takes into account the state of a voxel (active /inactive) without sacrificing the steerability.
The kernel construction follows the spherical harmonics weighted by the Clebsch-Gordan coefficients and radially weighted with a Gaussian.
The paper contains an extensive experimentation, comparing several 3D convolutions, and applying the method on 6dof instance pose, tracking and simultaneous class pose and size.




**Limitations And Societal Impact:**

Limitations explained in Conclusion.

**Main Review:**

This paper develops a new efficient scheme for steerable convolution on volumetric data. Volumetric convolutions are very expensive and any way to accelerate them has tremendous practical impact.

The paper takes the two main ideas of 3D steerability [24] and sparsified convolutions [9] to design a tensor-based sparsified convolution that is steerable in SO(3). The authors provide all details of the novel design. The reader needs some background reading on steerability and to understand the general steerability constraint (4) that is applicable for any dimension of the input and the output (the formula is much simpler for scalar fields).

The active sites (voxels) are stored in a hash table $H$ updated by the active/inactive state. This is a very elegant design indeed and $H$ together with the feature vector $F$ constitute the vector that is updated from layer to layer.

The paper is one of the few, might be next to Esteves ICML 2019 (equivariant embeddings that are used for relative SO(3) computations)  that contains extensive experimentation on non-classification tasks (no MNIST, Cifar10 etc) but standard 6dof object pose benchmarks like LineMOD and REAL275.

These applications alone could be a paper in a mainstream vision conference.

For instance-based 6dof pose the authors segment the volume creating the initial hash table and start with the vector (R,G,B,1) as feature input. After several layers, features are converted into point coordinates that are regressed to compute the pose.
At this point, the reader has really to go to the appendix to understand the loss functions. While ground-truth poses are used the loss functions are ICP-like. Please move the losses to the main text. The Appendix is highly informative and like the paper itself a delight to read.

The method outperforms all methods on pose and size and comparable or better to DenseFusion of G2L on instance-based 6dof pose estimation.

POSTREBUTTAL: I read all reviews and the authors' responses and I stay with my positive mark: The combination of steerability and sparisty is not a simple integration, and the experiments on pose are novel in the context of steerability. The reader can learn a lot by reading the paper.

**Time Spent Reviewing:**

3

---

> ### Author Response · Authors · 2021-08-10
> **Authors' Response**
>
> We highly appreciate the reviewer’s comments. We agree with the reviewer and will include essential designs about applications of the proposed SS-Conv, including loss functions, into the main text.

---

### Decision · Program_Chairs · 2021-09-28

**Decision:**

Accept (Poster)

**Comment:**

Initially, two of the reviewers expressed concerns about the paper and ranked the paper below acceptance. As the ensuing rebuttal managed to successfully address most of reviewer’s concerns, ACs and the majority of the reviewers agreed that this is a strong paper (highest rate was 9!)  and recommended acceptance. Authors are highly encouraged to address the key comments reported by reviewers as well as to implement all the improvements (as indicated by authors in the rebuttal) in the final camera-ready version.

**Consistency Experiment:**

NeurIPS has a long history of experimentation. In 2014, NeurIPS ran an experiment in which 10% of submissions were reviewed by two independent committees to quantify the randomness in the review process. This year, we repeated a variant of this experiment to see how the quality of the review process has changed over time.  This paper was part of the experiment and was therefore assigned to two committees (consisting of reviewers, an Area Chair, and a Senior Area Chair) that reached independent decisions.  If both committees made the same recommendation, this recommendation was followed. If a single committee recommended acceptance, the paper was accepted (with the exception of a few cases in which the other committee identified what we considered a fatal flaw, e.g., an error in a key result).

Both committees reached the same decision: **Accept (Poster)**

The other committee assigned to the paper recommended **Accept (Poster)**.  You can find the other set of reviews, along with any follow up discussion with the authors here:
https://openreview.net/forum?id=Fa-w-10s7YQ